# The Bioactive Value of *Tamarix gallica* Honey from Different Geographical Origins

**DOI:** 10.3390/insects14040319

**Published:** 2023-03-27

**Authors:** Ahmed G. Hegazi, Fayez M. Al Guthami, Mohamed F. A. Ramadan, Ahmed F. M. Al Gethami, A. Morrie Craig, Hesham R. El-Seedi, Inmaculada Rodríguez, Salud Serrano

**Affiliations:** 1Department of Zoonotic Diseases, National Research Centre, Dokki, Giza 12622, Egypt; 2Al Guthami Company, Makkah 24211, Saudi Arabia; 3Pesticide Analysis Research Department, Central Agric. Pesticides Lab., Agric. Res. Center, Giza 12611, Egypt; 4Alnahalaljwal Foundation, Makkah 21926, Saudi Arabia; 5College of Veterinary Medicine, Oregon State University, Corvallis, OR 97331, USA; 6International Research Center for Food Nutrition and Safety, Jiangsu University, Zhenjiang 212013, China; 7International Joint Research Laboratory of Intelligent Agriculture and Agri-Products Processing, Jiangsu Education Department, Jiangsu University, Nanjing 210024, China; 8Department of Chemistry, Faculty of Science, Menoufia University, Shebin El-Koom 32512, Egypt; 9Department of Food Science and Technology, Faculty of Veterinary Medicine, University of Córdoba, 14071 Córdoba, Spain

**Keywords:** antioxidant activity, antibacterial activity, melissopalynological analysis, physicochemical analysis, *Tamarix gallica* honey

## Abstract

**Simple Summary:**

The current study was conducted to characterize *Tamarix gallica* honey in terms of its antimicrobial and antioxidant activities, melissopalynological analysis, physicochemical and biochemical properties, and total phenolic and flavonoid contents. For this purpose, *Tamarix gallica* honey samples from Saudi Arabia, Libya, and Egypt were collected and analyzed. Our results reveal the antimicrobial and antioxidant capacities of this type of honey. *Tamarix gallica* honey could be considered for therapeutical, food manufacturing, or nutraceutical purposes.

**Abstract:**

This study was conducted to assess the bioactive value of *Tamarix gallica* honey samples collected from three countries. In total, 150 *Tamarix gallica* honey samples from Saudi Arabia (50), Libya (50), and Egypt (50) were collected and compared, based on the results of the melissopalynological analysis, their physicochemical attributes, antioxidant and antimicrobial activities, and biochemical properties, together with their total phenolic and total flavonoid contents. Depending on the geographical origin, we observed different levels of growth suppression for six resistant bacterial strains. The pathogenic microorganisms tested in this study were *Staphylococcus aureus, Streptococcus mutans, Klebsiella pneumoniae, Escherichia coli, Proteus vulgaris,* and *Pseudomonas aeruginosa.* There was a strong correlation between the polyphenol and flavonoid contents, as well as significant (*p* < 0.05) radical scavenging activities. The melissopalynological analysis and physicochemical properties complied with the recommendation of the Gulf and Egyptian Technical Regulations on honey, as well as the Codex Alimentarius of the World Health Organization and the European Union Normative related to honey quality. It was concluded that *Tamarix gallica* honey from the three countries has the capacity to suppress pathogenic bacterial growth and has significant radical scavenging activities. Moreover, these findings suggest that *Tamarix gallica* honey may be considered as an interesting source of antimicrobial compounds and antioxidants for therapeutical and nutraceutical industries or for food manufacturers.

## 1. Introduction

The drug resistance of some bacterial strains has provoked research into natural antimicrobial agents and plants, which could be the source of novel remedies [1]. Throughout history, natural products (particularly, honey) have been commonly used in nutrition, prevention, and as therapeutical agents. *Apis mellifera* worker honeybees collect nectar from different plants and produce honey with great diversity depending on the different botanical sources [2,3,4]. Thus, honey quality depends on different factors such as the botanical source, the plant chemical composition, the weather conditions, the soil mineral composition, and the geographical origin [5].

Honeybees produce important medicinal products, such as bee venom, royal jelly, honey, propolis, and beeswax. Honey is the most appreciated and widely used product [6,7]. Promising antimicrobial and antioxidant properties of honey can be applied in prophylaxis and therapeutical use for many diseases. Natural pure honey is well documented to contain novel antimicrobial compounds [8,9,10,11,12,13,14,15].

The antibiotic resistance of some bacterial strains has attracted research on natural honey as an antimicrobial agent [16]. The *Tamarix gallica* plant is indigenous to Saudi Arabia and the Sinai Peninsula, and it is common around the Mediterranean region. It is a tree or shrub halophyte from coastal regions and deserts, a relatively long-living plant that can tolerate a wide range of environmental conditions and resist abiotic stresses such as salt, high temperature, and drought stresses. *Tamarix* species are ornamental bushes or trees, known for their feathery foliage, mostly evergreen with pink or white blossoms [17]. Depending on the species, they may reach a large tree size, but most invasive species are multi-stemmed shrubs of less than 8 m that can grow 3 to 4 m in a growing season [18]. The leaves are alternate, sessile, small, punctate, and scalelike with salt-secreting glands and are self-pruning during drought periods [19]. The leaves of the plant are astringent and diuretic, used as an external compress for wounds to stop bleeding and as a laxative, an astringent, an antidiarrheal, and an antidysentery. The major chemical constituents of *Tamarix* are tamarixin, tamarixetin, troupin, 4-methylcoumarin, 3, 3′-di-0-methylellagic acid, and quercitol (methyl ester) [20]. Numerous polyphenols are also present, including anthocyanins, tannins, flavanones, isoflavones, resveratrol, and ellagic acid.

This genus is used in traditional medicine as a perspiration stimulant, an aperitif, a diuretic, and an astringent, for its antimicrobial activities and bioactive molecules [17]. It is also used in the treatment of various liver disorders due to its anti-inflammatory and antidiarrheic properties. Knowledge of the physical and chemical properties of *Tamarix gallica* honey is limited [21].

In general, honey authenticity is measured by different international standards, such as the *Codex Alimentarius* Standard, but this authenticity depends mainly on the geographical and botanical origins. As not much information about *Tamarix gallica* honey exists, the aim of this study was to assess the antimicrobial and antioxidant properties, as well as the total phenolic and total flavonoid contents, of *Tamarix gallica* honey from Saudi Arabia, Libya, and Egypt.

## 2. Materials and Methods

### 2.1. Materials

The reagents and chemicals used were of analytical grade and purchased from Sigma-Aldrich (St. Louis, MO, USA).

### 2.2. Honey Samples

In total, 150 fresh *Tamarix gallica* honey samples (1 kg each) were collected from Saudi Arabia (50), Libya (50), and Egypt (50) during the 2021 harvest. Each honey sample was collected in a sterile universal glass container and kept at 2–8 °C until tested. The melissopalynological analysis corroborated its botanical authenticity as *Tamarix gallica* honey, which meant that the pollen content of this specific floral source was at least 55% [22,23,24].

### 2.3. Bacterial Strains

Six antibiotic-resistant bacterial strains (Gram-positive and Gram-negative) were used in this investigation to determine the antibacterial activities of the honey: *Staphylococcus aureus* (ATCC 25923), *Streptococcus mutans* (1815T), *Proteus vulgaris* (ATCC 13315), *Pseudomonas aeruginosa* (ATCC 27853), *Klebsiella pneumoniae* (ATCC 27736), and *Escherichia coli* (ATCC 35218). The identification and susceptibility patterns of all the clinical isolates were performed using the VITEK 2 compact system of the Department of Zoonotic Diseases, National Research Centre (Cairo, Egypt), which kindly provided and maintained these bacterial strains. Muller Hinton Agar (MHA) was used to subculture the bacterial strains. Then, the incubation was carried out at 37 °C overnight. A homogeneous suspension was obtained from a single colony of the tested microorganism using a sterile loop and inoculating it in 3 mL of Muller Hinton Broth (Sigma-Aldrich, St. Louis, MO, USA). To provide 0.5 McFarland  =  1 – 2 × CFU/mL, the suspension was standardized using a calibrated VITEK 2 DENSICHEK (BioMérieux, Inc., Marcy l′Etoile, France) [25,26].

### 2.4. Disc Diffusion Method

The measure of both the growth and the inhibition of the control bacterial strains mixed with honey was carried out using the disc diffusion method. This method was performed using prepared discs of approximately 6 mm diameter (Whatman filter paper no. 1). They were sterilized in a hot air oven according to the Clinical and Laboratory Standards Institute (CLSI) guidelines [27] and spotted with 0.2 mg of *Tamarix gallica* honey. The preparation for each bacterial strain suspension was carried out by inoculating the fresh stock culture into a tube containing 10 mL of Muller Hinton Broth (Sigma-Aldrich company) and then incubated aerobically at 37 °C for 24 h. A 0.5 McFarland turbidity standard (5 × 10^7^ cells/mL) was used to adjust the bacterial suspension, followed by a further dilution to obtain 5 × 10^6^ cells/mL. For all the dilution steps, physiological saline PBS pH 7.2 was used under aseptic conditions. These bacterial strains were enriched on selective broth for bacterial propagation. A separate tube containing 40 µL of 21.30% honey concentration was mixed with 0.20 µL/10 mL from the enriched broth of each propagated *S. aureus*, *S. mutans*, *K. pneumoniae*, *E. coli*, *Proteus vulgaris*, and *P. aeruginosa*, and further incubated at 37 °C for 24 h. The calculation of the mean values of inhibition was carried out from the triplicate readings in each test. The determination for the evaluations of the antibacterial activity of the different honey dilutions was performed according to Hegazi et al. [15].

### 2.5. Determination of the Minimum Inhibitory Concentration (MIC)

The MIC of different samples of *Tamarix gallica* honey was determined by a two-fold serial dilution method. Concentrations of 50, 25, 12.50, 6.25, 3.12, 1.56, and 0.78 mg/mL and 390, 195, and 97 µg/mL were performed from a serial dilution of 100 mg/mL. Briefly, 100 µL of the varying sample concentrations was added separately to the test tubes containing 9 mL of the standardized suspension of the tested bacteria (10^8^ CFU/mL). Control tests with the studied organisms were performed using distilled water instead of honey. The test tubes were incubated at 37 °C for 24 h. The lowest concentration of these samples with no visible growth was taken as the MIC [28,29].

### 2.6. Detection of Total Phenolic Content (TPC)

The total phenolic content (TPC) was determined using Folin–Ciocalteu reagent, according to the method described by [30,31]. A honey solution of 0.5 mL was mixed with 2.5 mL Folin–Ciocalteu reagent (2N) and incubated for 5 min. Subsequently, 2 mL of sodium carbonate solution (75 g/L) was added and incubated for 2 h at 25 °C. After incubation, the absorbance was measured at a wavelength of 765 nm using a UV–Visible spectrophotometer (Perkin-Elmer Lambda 25, Waltham, MA, USA). For the calibration curve, a standard of gallic acid (0–1000 mg/L) was used. The mean value for the triplicate assays of the TPC was reported and expressed as milligrams of gallic acid equivalent (GAE) per gram of honey [32].

### 2.7. Determination of Total Flavonoid Content (TFC)

The total flavonoid content (TFC) was determined using a volume of 5 mL of diluted honey with a 0.1 g/mL concentration. This solution was mixed with 5 mL of 2% aluminum chloride (AlCl_3_). The mixture was then incubated for 10 min at 25 °C. The absorbance of the formed complex was measured at 415 nm using a UV–Visible spectrophotometer. The standard chemical for the calibration curve preparation was rutin with a concentration of 0–100 mg/L. The mean value of the triplicate assays of the TFC was reported and expressed as milligrams of rutin equivalent (RE) per gram of honey [32,33].

### 2.8. Antioxidant Assay to Determine the DPPH Scavenging Activity

The DPPH scavenging of the honey samples was determined by carrying out an antioxidant assay. This test is based on the reduction in the purple DPPH radical using an oxidizing antioxidant. The scavenging effects of vitamin C and caffeic acid corresponded to the quenching intensity of 1.1-diphenyl-2-picrylhydrazyl (DPPH), as carried out by [31,34]. A wavelength of 520 nm was set to measure the reduction in the purple DPPH radical.

### 2.9. Melissopalynological and Physicochemical Analysis

The quantitative and qualitative analyses of the pollen grains present in the honey were carried out by the detection of microscopic specimens [35] and based on the percentage of the pollen grain type that was the most dominant [36]. To classify honey as unifloral, this percentage of pollen type must show the highest counting of grains in the honey sediment, and the honey can be named after the plant these pollen grains come from [37,38].

The melissopalynological and physicochemical characterizations were determined. The sedimentation technique was used for the determination of pollen content as described by [24,36]. The water content [39], water-soluble solids [40], pH, total acidity, and electrical conductivity were also analyzed [41]. The analyses of sugar content, glucose, fructose, fructose/glucose ratio, fructose plus glucose %, and sucrose were performed by HPLC-DAD according to official methods [42]. The diastase activity was determined photometrically by the Phadebas method [43]. The results are expressed as a diastase number (DN) in Gothe or Schade units. One unit corresponds to the enzyme activity of 1 g of honey, which can hydrolyze 0.01 g of starch in 1 h at 40 °C and pH 5.2. The calculation modified by Bogdanov et al. [44], DN = 28.2 × ΔA_620_ + 2.64, was used. The determination of the hydroxymethylfurfural was made according to the Winkler method [45]. The results are expressed in HMF milligrams per kg of honey.

### 2.10. Statistical Analysis

SPSS Ver. 21 (IBM, New York, NY, USA) software for the statistical analysis was used on the triplicate results. The comparison between and within the tested groups was applied using one-way ANOVA. The mean ± standard error (SE) was used for all data, and a *p*-value less than 0.05 was considered significant.

## 3. Results

The antibacterial activities of *Tamarix gallica* honey from the three countries were evaluated according to the zone of growth inhibition. The tested bacterial strains showed growth suppression in all the honey types (Table 1). Egyptian *Tamarix gallica* honey showed the highest zones of inhibition of 23.10 ± 0.38 mm, 29.33 ± 0.64 mm, 24.02 ± 0.34 mm, and 22.00 ± 0.58 mm against *Staphylococcus aureus, Streptococcus mutans*, *Klebsiella pneumoniae,* and *Pseudomonas aeruginosa,* respectively. Penicillin, Oxacillin, and Clindamycin showed effective antibacterial activity against the tested bacteria (Table 1).

The MIC tests are shown in Table 2. All the honey showed antibacterial potential against *S. aureus, S. mutans, K. pneumoniae*, and *E. coli*. However, the Libyan *Tamarix gallica* honey showed the strongest antibacterial potential against *Pseudomonas aeruginosa* and the Egyptian honey against *Proteus vulgaris*. The therapeutical antibiotics Penicillin, Oxacillin, and Clindamycin showed the highest MIC activity against all the tested bacteria (Table 2).

The total phenolic (mg GAE/100 g honey), total flavonoid (mg RE/100 g honey), and DPPH (mg AAE/100 g honey) contents are shown in Table 3. The highest level of total phenolics of the *Tamarix gallica* honey was observed in the honey from Egypt at 142.29 ± 15.32 (mg GAE/100 g honey), with the lowest level observed in the Saudi Arabian honey. The highest level of total flavonoids (mg RE/100 g honey) was detected in the honey from Saudi Arabia with a value of 83.1 ± 18.33 (mg RE/100 g honey). Moreover, the highest value of DPPH was detected in Egyptian honey.

The melissopalynological analysis of the *Tamarix gallica* honey from the different locations revealed that not only the specific botanical source giving the nectar but also other pollen grain types from different botanical sources were present in the pollen spectra. The pollen content and types obtained varied depending on the geographical origin of the sample (Figure 1 and Table 4).

The results of the physicochemical parameters (Table 5) revealed that the *Tamarix gallica* honey samples were comparable in water content, which ranged from 13.2 ± 0.58% (Libya) to 16.15 ± 0.11% (Egypt). The pH ranged from 4.20 ± 0.58 (Saudi Arabia) to 4.26 ± 0.53 (Egypt). The total acidity also varied from 13 ± 0.58 (Libya) to 21.5 ± 0.61 meq/l (Egypt). The electrical conductivity was 0.4 ± 0.003 (Libya) to 0.51 ± 0.004 mS/cm (Saudi Arabia). The insoluble solids ranged from 0.064 ± 0.006% (Saudi Arabia) to 0.075 ± 0.003% (Libya). The glucose, fructose, sucrose, and diastase activity levels in the different *Tamarix gallica* honey samples are shown in Table 5. The percentages of sugars were similar for the three geographical origins. The Egyptian honey had the highest value for diastase activity (28.15 ± 0.47 DN) and the lowest HMF (5.95 ± 0.55 mg/kg).

## 4. Discussion

The antibacterial activities of all the honey in this study were similar, which may be attributed to the narrow ranges of their total phenolic and flavonoid contents. There is a positive correlation between the TPC and the antibacterial activity of honey [46], which is suggested to be due to the inhibition in the virulence factors of the pathogen [47]. On the other hand, Makarewicz et al. [48] found a weak correlation between the antioxidant and antimicrobial activities of some Polish commercial honeys. 

Honey has strong antibacterial activity due to the following factors: the phenolic compounds, the low pH, the high osmolarity, the hydrogen peroxide produced by glucose oxidase enzyme [13,14,49], and the presence of lysozyme, methylglyoxal, bee peptides, and high sugar contents [50]. The *Tamarix gallica* honey studied in this work showed different inhibition activities against the tested bacteria based on its origin, possibly due to its major phenolics and flavonoids: syringic acid, isoquercetin, gallic acid, ρ-coumaric, and catechin [51]. The presented results were in accordance with the research of several authors [50,52,53,54,55] stating that both Gram-positive and Gram-negative pathogenic bacteria are susceptible to honey. Other authors found the antimicrobial activity of honey to be directly related to its botanical origin and phenolic compounds, including mainly flavonoids and phenolic acids [16].

The total phenolic content ranged from 129.89 ± 10.66 to 142.29 ± 15.32 mg GAE/100 g honey, which was higher than the results found for honey from different countries, namely India [56] (47–98 mg GAE/100 g honey), Poland [57] (71.7 to 202.6 μg/g honey), Argentina [31] (18.730–107.213 mg GAE/100 g honey), Burkina Faso [58] (32.59–114.75 mg GAE/100 g honey), Portugal [55] (30.87 to 87.27 mg GAE/100 g), and Romania [59] (2–125 mg GAE/100 g honey). The total phenolic contents in different honey types have been reported [60,61,62,63]. The variability was associated with the floral origin of the honey, and multifloral honey was found to have higher phenolic contents than unifloral honey. The phenolic compounds, especially flavonoids in honey, have been reported to have antiviral, antimicrobial, antifungal, antioxidant, and anti-inflammatory activities [64].

Moreover, the flavonoid content depends mainly on the geographical and botanical origin. The total flavonoids in the Saudi Arabian and Egyptian *Tamarix gallica* honeys (83.1 ± 18.33 and 75.5 ± 13.47 mg RE/100 g honey, respectively) were greater than (*p* < 0.05) the Libyan ones (63.6 ± 11.13 mg RE/100 g honey). Similar results were obtained in the analysis of three unifloral honey types from Portugal, which showed that the darker the honey, the richer the phenolic content [65].

The antioxidant activity of the *Tamarix gallica* honey samples was determined by the DPPH radical scavenging method. The *Tamarix gallica* honey from Egypt (173.80 ± 10.51 mg AAE/100 g honey) and Saudi Arabia (153.30 ± 15.18 mg AAE/100 g honey) presented the highest levels of DPPH radical scavenging activities (*p* < 0.05), compared with the antioxidant activities of the Libyan honey samples. Similar results were reported in other studies [66,67,68]. Alves et al. [69] observed a positive correlation between the phenolic concentration, the antioxidant capacity, and the color of the honey. The volatile compounds came from the diverse floral sources [70,71] and were considered to be responsible for the most potent biological activity and of medical importance.

A long honey shelf life depends on its low water content, while high moisture in honey during storage promotes the process of fermentation. The water content of the investigated *Tamarix gallica* honey samples complied with the accepted range. According to the *Codex Alimentarius,* the water content of honey should not exceed 20%. The relative humidity and temperature affect the water content during honey production by honeybees [72]. 

With the aim of assessing the authenticity and the overall quality of the honey, the sugar content is determined [73]. The sugar analysis of honey is a good indicator of artificial (sugar solution) or natural (nectar) feeding of honeybees. The use of sugar solution to feed honeybees is showed when the glucose content in honey is much higher than its fructose content [74]. Our results revealed that the sum of the glucose and fructose, which measures the content of the reducing sugars in honey, was within the accepted range to prove the standardization and the authenticity of the honey, as observed by Aljohar et al. [7]. Fructose is the most dominant sugar in honey samples [75], and the ratio of fructose to glucose (F/G) indicates the natural feeding of honeybees [76]. The sucrose content in all honey samples did not exceed 5% in this study, which is the accepted level to prove the authenticity of honey, as observed by [77]. The diastase activity and hydroxymethylfurfural content are also important parameters used to prove the freshness of honey [78,79]. Diastase activity depends on many factors, such as the physiological period of the colony, the age of the bees, the nectar collection season, the amount of nectar, and its sugar content. The *Codex Alimentarius* of the World Health Organization, the European Union Quality Normative on honey, and the Gulf Technical Regulation on honey (GSO 147:2008-Standards Store-GCC Standardization Organization EOSC, 2005) recommend that the maximum level for the HMF content should not exceed 40 mg/kg; in countries with tropical temperatures, the HMF content should not exceed 80 mg/kg. In this study, all the examined *Tamarix gallica* honey samples complied with the accepted limit for the HMF content and diastase number, which are indicators of the authenticity and freshness of honey. All the *Tamarix gallica* honey samples were found to be within the accepted range of acidity. The honey acidity is related to the presence of organic acids, particularly gluconic acid, which was found to affect the honey flavor, texture, shelf life, and stability [80].

## 5. Conclusions

We have concluded that *Tamarix gallica* honey from Egypt, Saudi Arabia, and Libya has the capacity to suppress pathogenic bacterial growth and has significant free radical scavenging activities. Moreover, these findings suggest that *Tamarix gallica* honey may be considered as an interesting source of antimicrobial compounds and antioxidants for therapeutical or nutraceutical industries and for food manufacturers. The physicochemical and melissopalynological characterization was carried out, describing this type of unifloral honey for the first time.

## Figures and Tables

**Figure 1 insects-14-00319-f001:**

Images of pollen grains from microscopic preparations of the *Tamarix gallica* honey from different geographical origins.

**Table 1 insects-14-00319-t001:** The inhibition zone (in mm) of *Tamarix gallica* honey against various pathogenic microorganisms by the disc diffusion method (mean values ± SE). Each test was run in triplicate.

Antibacterial Activity	Gram-Positive		Gram-Negative			
Honey Origin	*Staphylococcus aureus*	*Streptococcus mutans*	*Klebsiella pneumoniae*	*Escherichia coli*	*Proteus vulgaris*	*Pseudomonas aeruginosa*
Saudi Arabia	21.33 ± 0.88	18.33 ± 0.88	22.05 ± 0.68	15.33 ± 0.33	21.50 ± 0.30	21.00 ± 1.15
Libya	16.01 ± 0.58	16.33 ± 0.88	11.03 ± 0.56	18.05 ± 0.58	16.06 ± 0.31	19.67 ± 0.33
Egypt	23.10 ± 0.38	29.33 ± 0.64	24.02 ± 0.34	19.16 ± 0.60	20.01 ± 0.34	22.00 ± 0.58
Penicillin	53.02 ± 0.64	37.00 ± 0.64	33.12 ± 0.38	32.35 ± 0.12	41.14 ± 0.21	37.21 ± 0.28
Oxacillin	46. 21 ± 0.33	33.02 ± 0.36	50.01 ± 0.13	26.29 ± 0.18	38.22 ± 0.29	39.33 ± 0.16
Clindamycin	50. 55 ± 0.31	34.01 ± 0.22	44.41 ± 0.23	29.11 ± 0.41	45.14 ± 0.30	42.17 ± 0.19

**Table 2 insects-14-00319-t002:** The minimal inhibitory concentration (MIC *), in mg/mL, of *Tamarix gallica* honey from Saudi Arabia (n = 50), Egypt (n = 50), and Libya (n = 50) against the antibiotic-resistant bacterial strains.

**Gram-positive**										
	**50 mg/mL**	**25 mg/mL**	**12.5 mg/mL**	**6.25 mg/mL**	**3.12 mg/mL**	**1.56 mg/mL**	**0.78 mg/mL**	**390 µg/mL**	**195 µg/mL**	**97 µg/mL**
*Staphylococcus aureus*										
Saudi Arabia	0.08	0.15	0.2	0.25	0.3	0.4	0.4	0.4	0.4	0.4
Libya	0.07	0.2	0.25	0.3	0.3	0.4	0.4	0.4	0.4	0.4
Egypt	0.05	0.15	0.175	0.2	0.3	0.4	0.4	0.4	0.4	0.4
*Streptococcus mutans*										
Saudi Arabia	0.09	0.15	0.2	0.25	0.27	0.4	0.4	0.4	0.4	0.4
Libya	0.09	0.15	0.25	0.3	0.3	0.4	0.4	0.4	0.4	0.4
Egypt	0.08	0.1	0.175	0.2	0.3	0.4	0.4	0.4	0.4	0.4
**Gram-negative**										
	**50 mg/mL**	**25 mg/mL**	**12.5 mg/mL**	**6.25 mg/mL**	**3.12 mg/mL**	**1.56 mg/mL**	**0.78 mg/mL**	**390 µg/mL**	**195 µg/mL**	**97 µg/mL**
*Klebsiella pneumoniae*										
Saudi Arabia	0.1	0.12	0.2	0.25	0.27	0.4	0.4	0.4	0.4	0.4
Libya	0.1	0.2	0.25	0.3	0.35	0.4	0.4	0.4	0.4	0.4
Egypt	0.1	0.1	0.175	0.2	0.3	0.4	0.4	0.4	0.4	0.4
*Escherichia coli*										
Saudi Arabia	0.09	0.15	0.2	0.25	0.27	0.4	0.4	0.4	0.4	0.4
Libya	0.09	0.15	0.25	0.3	0.3	0.4	0.4	0.4	0.4	0.4
Egypt	0.08	0.1	0.175	0.2	0.3	0.4	0.4	0.4	0.4	0.4
*Proteus vulgaris*										
Saudi Arabia	0.09	0.1	0.17	0.25	0.25	0.2	0.2	0.2	0.2	0.2
Libya	0.09	0.15	0.25	0.3	0.3	0.4	0.4	0.4	0.4	0.4
Egypt	0.08	0.1	0.17	0.2	0.3	0.1	0.1	0.1	0.1	0.1
*Pseudomonas aeruginosa*										
Saudi Arabia	0.08	0.1	0.15	0.2	0.25	0.4	0.4	0.4	0.4	0.4
Libya	0.09	0.1	0.15	0.12	0.3	0.1	0.1	0.1	0.1	0.1
Egypt	0.08	0.1	0.12	0.2	0.3	0.4	0.4	0.4	0.4	0.4
**Drugs for positive control for growth inhibition**										
	*Staphylococcus aureus*		*Streptococcus mutans*		*Klebsiella pneumoniae*	*Escherichia coli*		*Proteus vulgaris*		*Pseudomonas aeruginosa*
Penicillin	8 × 10^−6^		1.6 × 10^−6^		6.4 × 10^−7^	8 × 10^−6^		1.28 × 10^−5^		1.6 × 10^−7^
Oxacillin	1.6 × 10^−8^		3.2 × 10^−8^		1.28 × 10^−7^	1.6 × 10^−7^		8 × 10^−6^		6.4 × 10^−8^
Clindamycin	8 × 10^−7^		1.28 × 10^−6^		6.4 × 10^−8^	8 × 10^−7^		1.6 × 10^−6^		1.6 × 10^−7^

* MIC, concentration required for 99% bacteriostatic effect.

**Table 3 insects-14-00319-t003:** The total phenolics, total flavonoids, and DPPH of the *Tamarix gallica* honey (mean values ± SE).

Honey	Samples (n)	Total Phenolics (mg GAE/100 g Honey)	Total Flavonoids (mg RE/100 g Honey)	DPPH (mg AAE/100 g Honey)
Saudi Arabia	50	129.89 ± 10.66	83.1 ± 18.33	153.30 ± 15.18
Libya	50	134.11 ± 13.30	63.6 ± 11.13	101.00 ± 11.82
Egypt	50	142.29 ± 15.32	75.5 ± 13.47	173.80 ± 10.51

**Table 4 insects-14-00319-t004:** Pollen analysis of the *Tamarix gallica* honey from different origins.

Honey Origin	Botanical Family	Botanical Species	Frequency of Occurrence *
Saudi Arabia	Rhamnaceae Mimosaceae Tamaricaceae	*Ziziphus jujuba* *Acacia asak* *Tamarix gallica*	++ + ++++
Libya	Rhamnaceae Tamaricaceae Tamaricaceae	*Ziziphus jujuba* *Tamarix atlantis* *Tamarix gallica*	++ ++ ++++
Egypt	Rhamnaceae Poaceae Tamaricaceae Tamaricaceae Brassicaceae	*Ziziphus lotus* *Oryza meyeriana* *Tamarix nilotica* *Tamarix gallica* *Brassica tournefortii Gouan*	++ ++ ++ ++++ +

* ++++: more than 70%; ++: 60%; +: 50%.

**Table 5 insects-14-00319-t005:** The physicochemical parameters of the *Tamarix gallica* honey from Saudi Arabia (n = 50), Egypt (n = 50), and Libya (n = 50) (mean values ± SE).

Physicochemical Parameters	Honey Origin		
	Saudi	Libya	Egypt
**Moisture (%)**	14.94 ± 0.88	13.2 ± 0.58	16.15 ± 0.11
**Fructose (%)**	40.94 ± 0.68	39.33 ± 0.48	39.74 ± 0.33
**Glucose (%)**	35.64 ± 0.44	35.13 ± 1.48	35.64 ± 0.58
**Fructose/glucose ratio**	1.14	1.11	1.11
**Fructose + Glucose (%)**	76.57 ± 1.85	74.46 ± 3.05	75.38 ± 2.66
**Sucrose (%)**	1.28 ± 0.30	1.75 ± 0.22	1.75 ± 0.32
**HMF (mg/kg)**	22.36 ± 0.10	20.25 ± 0.45	5.95 ± 0.55
**Acidity (meq/l)**	19.6 ± 1.61	13 ± 0.58	21.5 ± 0.61
**Diastase (DN)**	18.9 ± 0.38	10.25 ± 0.64	28.15 ± 0.47
**Electrical conductivity (mS/cm)**	0.51 ± 0.004	0.4 ± 0.003	0.47 ± 0.0022
**Water-insoluble solids content (%)**	0.064 ± 0.006	0.075 ± 0.003	0.065 ± 0.004
**pH**	4.20 ± 0.58	4.25 ± 0.27	4.26 ± 0.53

## Data Availability

The data supporting reported results can be found in the manuscript.

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
