# Peer review of "The Bioactive Value of Tamarix gallica Honey from Different Geographical Origins"

_insects, 2023, doi:10.3390/insects14040319_

Round 1

Reviewer 1 Report

Dear Author, the study is interesting but there are at least two problems in this paper: 1) Data presented in table 2 are not correctly shown and discussed. 2) some references are not pertinent with the text.

Here some minor corrections:

Line 33: …an interesting source of antimicrobial and antioxidants… antimicrobial is an adjective, so maybe a noun is missing: antimicrobial compounds?

Add space between words in lines 101 (thediameter) 106 (transferring 100 μl.Final concentrations) 109 (thedilution) 162 (andhoney) 188 (antibioticresistant)

Line 183: …effective antibacterial activity over the tested bacteria (Figure 1).: correct: (Table 1)

Line 200: …content is shown in Figure 3.: correct: in Table 3

Line 209:.. otdifferent botanical: different

Line 222: higher: highest

Line 228: … the narrow ranges of their total phenolic and flavonoid. Here as well a noun seems missing: maybe:… the narrow ranges of their total phenolic and flavonoid contents?

Comments:

Line 45: reference (2) is not pertinent. Reference (3) regards microbial resistance, it is not in the right context here.  

Some references in the section Materials and Methods as well seem to me not pertaining:

Line 115: the reference (29) seems to me not pertinent with the paper, dealing with Preparation of ace inhibitory peptides from mytilus coruscus hydrolysate using uniform design. It doesn’t seem to me that the Uniform design used in the mentioned paper is applied in present work.

Line 130: the reference (30) as well does not seem pertinent with present study: it deals with a completely different field of application of extract of the plant Tamarix gallica, not honey.  

Line 128, reference (32): A Hidden Pitfall in the Preparation of Agar Media Undermines Microorganism Cultivability. I did not find a connection with present work in this cited paper.

Results

Line 188: The results evidenced significant differences in MIC values against tested antibioticresistant strains… : the significant differences are not evident in the table or discussion.

From line 190 to 194: comment: if MIC is the: Minimal inhibitory concentration, and: concentration required for 99% bacteriostatic effect as reported in the Table 2, I do not understand why it is affirmed that:.. … Libyan Tamarix gallica honey showed the strongest antibacterial potential against all bacteria (MIC=0.4g/ml) except Pseudomonas aeruginosa (MIC=0.1g/ml) and so on.

0.1 and 0.2 g/ml are lower concentrations than 0.4 g/ml, so honeys with MIC of 0.2 and 0.1 have stronger antibacterial activity than those requiring 0.4 g/ml for the microbial inhibition: in fact, Penicillin, Oxacillin and Clindamycin show concentrations infinitively lower, to obtain the same result, having a very high antibacterial activity. Therefore, the results seem the opposite of what is stated in this section.

The concentrations (0.4; 0.2 etc) are not those shown in the Materials and Methods (line 107)

Therefore, in the discussion as well (line 239-240) it is affirmed that Gram-positive bacteria were more sensitive to the honey samples than Gram-negative ones. Instead, from the results it seems quite the opposite. Moreover, … The results showed significant differences in MIC values against tested antibiotic-resistant strains…(line 237) this is not evident from the table. Significance is not shown.

I suggest explaining better the method for MIC in the section Materials and Methods, make again the Table 2, adding the number of samples, calculate the significance and so on.  Make again the discussion as well, of course.

Reviewer 2 Report

The work of Hegazi et al. is the first report on the properties of Tamarix gallica honey. It is especially valuable to compare the same variety of honey from 3 countries. The work is interesting but requires numerous corrections before publication.

The manuscript requires extensive linguistic proofreading, preferably by a native speaker.

line 33: please add word compounds ("antimicrobial compounds")

Keywords: please separate "Melissopalynological and Physiochemical 35
analysis"

lines 42-43 - sentence needs rewriting

line 48 - listed bee products cannot be called "compounds", please correct to substances or products

line 50-51 - sentence needs rewriting

I suggest adding a brief botanical and phytochemical description of the plant to the Introduction

line 106 - unfinished sentence

line 124 - without a dot after the Celsius degree symbol and a space after the brackets

line 144 - AlCl3

line 169 - explain the methods used to determine diastase and HMF

Tables in Results section - the way of marking the statistical differences is unclear, please explain or replace with letters

In the description of the disc diffusion method, carvacrol was described as a positive control. Why this compound and where are the results obtained for it?

Table 2 - use exponential notation for MIC values for antibiotics

line 200 - results are in the table not figure

line 209 - what other examples of pollen were detected during the melispallinological analysis?

line 245: it should be: "phenolic compounds, including mainly flavonoids and phenolic acids"

In the discussion, please refer to the biological properties of Tamarix gallica, it may suggest what its bioactive components may affect the bioactivity of honey.

Please pay special attention to the formatting of the text and careful writing, e.g. insert spaces before the units, because their absence reduces the readability of the results.

Round 2

Reviewer 1 Report

Dear Author, the paper is improved, nevertheless some imperfections in the new version are added:

Line 17:…. physio chemical, and biochemical properties… in the original version it was physicochemical (like in line 27: physicochemical attributes) why did you change in “physio”?

Line 28:… tivities,, (2 commas)

Line 72: … tamarixin, tamarixetin,troupin, 4- methylcoumarin, 3, 3’-di-0-methylellagic acid and quercetol (methyllic ester)..

Comment: I never met the word “methyllic” before, so I searched online and found it only once, mentioned in a paper: https://www.researchgate.net/publication/306182825_Tamarix_gallica_For_traditional_uses_phytochemical_and_pharmacological_potentials#fullTextFileContent

where exactly the same words of your paper were used: “... constituents … tamarixin, tamarixetin, troupin, 4- methylcoumarin, 3, 3’-di-0-methylellagic acid and quercitol (methyllic ester)…” unfortunately, the error remains in your paper as well, it is: “methyl ester” instead.  Better adding the reference, moreover, since “an external compress…etc” as well is taken from that paper (I am not among the authors, anyway!).

Table 2: now the shown results seem correct; nevertheless, the table in present form is really almost impossible to read: you should try to put down another layout.

Therefore, I suggest a minor revision

Author Response

Answers to Reviewer 1:

Dear Author, the paper is improved, nevertheless some imperfections in the new version are added:

Authors thank the reviewer his/her revision.

Line 17:…. physio chemical, and biochemical properties… in the original version it was physicochemical (like in line 27: physicochemical attributes) why did you change in “physio”?

Thanks. It must be a mistake. Now it has been corrected.

Line 28:… tivities,, (2 commas)

Corrected.

Line 72: … tamarixin, tamarixetin,troupin, 4- methylcoumarin, 3, 3’-di-0-methylellagic acid and quercetol (methyllic ester)..

Comment: I never met the word “methyllic” before, so I searched online and found it only once, mentioned in a paper: https://www.researchgate.net/publication/306182825_Tamarix_gallica_For_traditional_uses_phytochemical_and_pharmacological_potentials#fullTextFileContent

where exactly the same words of your paper were used: “... constituents … tamarixin, tamarixetin, troupin, 4- methylcoumarin, 3, 3’-di-0-methylellagic acid and quercitol (methyllic ester)…” unfortunately, the error remains in your paper as well, it is: “methyl ester” instead.  Better adding the reference, moreover, since “an external compress…etc” as well is taken from that paper (I am not among the authors, anyway!).

Author thank the suggestion. Now, the term has been corrected and the reference added.

Table 2: now the shown results seem correct; nevertheless, the table in present form is really almost impossible to read: you should try to put down another layout.

The reviewer is right. This happens when track changes is used, and the table looks unconfigured. Now, another layout is presented.

Therefore, I suggest a minor revision

Reviewer 2 Report

Thanks to the authors for the corrections and responses to the review. Due to the improvements introduced, I recommend accepting the work for printing, after minor corrections:

- line 17: physicochemical, not 'physio chemical'

- Table 2 - please correct the formatting of the table, it is unreadable in its current form

Author Response

Answers to Reviewer 2

Thanks to the authors for the corrections and responses to the review. Due to the improvements introduced, I recommend accepting the work for printing, after minor corrections:

Authors thank the reviewer the improvement of the manuscript.

- line 17: physicochemical, not 'physio chemical'

Thanks. It has been corrected.

- Table 2 - please correct the formatting of the table, it is unreadable in its current form

The reviewer is right. This happens when track changes is used, and the table looks unconfigured. Now, another layout is presented.